# INTERPRETABLE WORD-LEVEL CONTEXT-BASED SENTIMENT ANALYSIS

## ABSTRACT

We propose an attention-based multiple instance classification model (AMIC) to conduct interpretable word-level sentiment analysis (SA) using only document sentiment labels. The word-level SA adds more interpretability compared to other models while maintaining competitive performance at the document level. Furthermore, we decompose our model into interpretable outputs that provide context weighting, indication of word neutrality, and negation. This structure provides insights on how context influences sentiment and the inner workings in the model's decision-making process. AMIC is built on a straightforward modeling framework (i.e., multiple instance classification model) which incorporates blocks of self-attention and positional encoded self-attention to achieve competitive prediction performance. The architecture is transparent yet effective at conducting interpretable SA. Model performance is reported on two document sentiment classification datasets, with extensive analysis of model interpretation.

## 1 INTRODUCTION

In this study, we propose a transparent and simple sentiment analysis (SA) architecture to tackle the black-box nature of transformer-based language models. It is capable of providing more interpretable results and gaining more insight into the sentiment decision-making process, which is often obscure in large language models. In particular, our approach combines word level transformer outputs in a manner that incentivizes each output to learn an interpretable property in SA like negation and word neutrality in the context of the entire document.

Although language models have demonstrated their effectiveness in SA, they are often criticized as being "black box" due to their complex structures and difficult interpretability. Especially for methods which focus on sentence-level or document-level analysis, the reasoning behind the final classification/prediction is often unclear. This lack of interpretability can undermine trust in the results and prevent applications of the models in areas where interpretability is as important as predictive accuracy (Petch et al., 2022). Therefore, there is a growing need for interpretable high-performance SA models that can provide greater transparency in the inference processes.

One way to conduct interpretable SA is to develop word-level context-based SA (WCSA) approaches. WCSA provides word-level sentiment measures while incorporating its local and global contexts, giving a more granular understanding of the key words or phrases that contribute to the overall sentiment of a text. Despite the advantages that WCSA can provide, most SA studies do not focus on individual words. Instead, they are conducted at higher levels such as at the sentence or document level. This is because assigning sentiment scores to individual words can be challenging due to the ambiguous and context-dependent nature of sentiment.

In this paper, we combine a straightforward modeling framework (i.e., multiple instance classification model) and transformer components (self-attention and self-attention with relative position representations) to produce a transparent/shallow yet effective WCSA architecture. The proposed algorithm is called the attention-based multiple instance classification (AMIC) model. The MIC modeling framework provides excellent transparency in the reasoning of the underlying processes. The self-attention components help ensure that AMIC can effectively process complex language patterns. Furthermore, the quantification of the global dependency and local dependency in contextual influence enables AMIC to effectively handle more delicate linguistic complexities such as negation,

sentiment intensity, and word neutrality. In short, AMIC harnesses the best of two worlds: the interpretability of a statistical modeling structure and the high predictive performance of transformer components.

## 1.1 RELATED WORK

Before the recent popularity of large language models, SA was conducted using statistical methods, which employ rigorous probability-based approaches to modeling text data (Medhat et al., 2014). For example, Tyagi & Sharma (2018) used logistic regression to detect hate speech in tweets. Naive Bayes models and support vector machine have been used to classify movie reviews as positive or negative (Das & Chen, 2007; Pang et al., 2002). These relatively simple statistical models are often considered interpretable and straightforward, as they assume clear relationships between the input text and the outcome, allowing for a transparent explanation of the model's predictions. However, most of the statistical SA models don't have a mechanism for incorporating document context effectively. Thus, situations where words or phrases may carry different sentiments/meanings depending on their contexts result in unreliable performances.

Deep neural networks in recent years have gained popularity due to their flexible modeling capability. One of the key advantages is their ability to capture document context, where the context of a word or sentence is dependent on surrounding words, which helps to elucidate its meaning (Mikolov et al., 2013b; Zhang & Wallace, 2015). For example, Kim (2014) first used CNNs for sentiment classification of movie reviews and showed that CNNs can effectively capture local contextual patterns. The long-short term memory (LSTM) model (Hochreiter & Schmidhuber, 1997) and its variant, the Bidirectional LSTM (BiLSTM) model (Zhang et al., 2015), have been successful in SA, especially when combined with CNNs (Minaee et al., 2019). One of the most successful transformer models is the Bidirectional Encoder Representations from Transformers (BERT) model, proposed by Devlin et al. (2019). With the self-attention mechanism, it can generate context-aware representations. Numerous works have developed transfer learning approaches from BERT that perform state-of-the-art in SA (Singh et al., 2021; Zhao & Yu, 2021; Wu & Ong, 2021). Moreover, recent work has also shown BERT is resilient to counterfactual augmentations (Kaushik et al., 2020).

## 2 MODEL COMPONENTS

### 2.1 MULTIPLE INSTANCE CLASSIFICATION

Multiple instance learning (MIL) is a form of weakly supervised learning where the classification (MIC) or prediction (MIP) task is performed on a set of labeled bags, each containing a collection of instances whose labels are often unobserved. Each individual instance is described by a set of covariates (or features). Instances in a bag contribute to the observed bag-level response (or label). MIL was first introduced by Dietterich et al. (1997) for drug activity prediction. The bag label is positive if at least one instance label is positive, and the bag label is negative if all instance labels are negative. The goal is to predict the label of a new bag. More details of MIL can be found in Carbonneau et al. (2018).

Ray & Page (2001) presented an approach based on primary instance, which assumes that the bag label is solely determined by the primary instances, while the non-primary instances carry little information on the bag label. Xiong (2022) followed this assumption and introduced a Bayesian MIC approach for cancer detection using T-cell receptor sequences. It is composed of two nested probit regression models, where the inner model predicts the primary instances and the outer model predicts bag labels based on the features of the primary instances identified by the inner model.

Note that the task of predicting the sentiment in text documents can be formulated as an MIC problem. Each text can be considered as a bag consisting of individual words as instances, where the features of these instances are represented by the corresponding word embeddings. Predicting the overall sentiment of text documents is equivalent to predicting the bag labels.

Specifically, we assume that words in text can be categorized either as sentiment words or as function words. Sentiment words are associated with a clear sentiment polarity, describing an emotion or experience that is either pleasant/desirable or unpleasant/undesirable. On the other hand, function words are words that are used to structure the sentence and convey meaning without sentiment

implications, such as most prepositions, conjunctions, articles, pronouns, auxiliary verbs, etc. With the MIC modeling framework, AMIC is able to recognize sentiment words, estimate sentiment score at the word level, determine the overall sentiment of a review by combining the sentiment scores of individual words, and provide interpretable results in SA.

## 2.2 SELF-ATTENTION WITH RELATIVE POSITION REPRESENTATIONS

The self-attention mechanism when incorporated in SA enables the model to assess the relationships between words in text. The connections and relationships between words are commonly referred to as dependencies (Nivre, 2005). Such dependencies can be categorized into two types: global dependency and local dependency. Global dependency refers to the long-range relationships between words across the entire text. Local dependency focuses on the immediate relationships between neighboring words within a small window of text, which includes dependencies such as the impact of negation words on sentiment. Capturing local dependencies can help a SA model to effectively utilize the subtle nuances and relationships among words that collectively contribute to the overall sentiment expressed in text.

The self-attention mechanism was originally designed to emphasize the relationships between words throughout the entire sentence, regardless of their specific positions. This characteristic makes it highly proficient in capturing global dependencies. However, its ignorance to the positions of words makes it incompetent in incorporating local dependencies. Without the knowledge of neighboring words, the self-attention mechanism, on its own, struggles to accurately discern sentiment shifts influenced by factors such as negation, word order, or proximity to other words. For example, self-attention can not distinguish the sentiment in the following two sentences with opposite sentiment, where the only difference is the placement of the word "not":

> Sentence I: The service of the restaurant is good, the overall experience is **not** bad.
> Sentence II: The service of the restaurant is **not** good, the overall experience is bad.

To incorporate positional information into self-attention, researchers have proposed different approaches (Vaswani et al., 2017; Shaw et al., 2018; Bilan & Roth, 2018; Chen et al., 2021; Huang et al., 2018). The method, introduced by Shaw et al. (2018), is known as self-attention with relative positional representations. The approach explicitly captures the relative positional information of words in a text by introducing relative positional embeddings, which encode both the position and direction between words. To incorporate the relative positional information in self-attention, two separate sets of relative positional embeddings are learned in the Key and Value vectors, respectively. Then the updated Key and Value vectors with positional awareness are calculated as the original self-attention Key and Value vectors plus their respective relative positional embeddings. The incorporation of relative positional information enables self-attention to consider the positional relationships between words and incorporate them in the computation of attention weights.

## 3 APPROACH

The AMIC architecture is designed with word level terms that each have specific interpretable behaviors such as 1) identifying whether a word in a text is a sentiment word or a functional word (i.e., the neutrality of the word), 2) computing the context-independent sentiment for each word, which represents a word's intrinsic context-free sentiment, 3) computing the contextual global dependency and local dependency for each word respectively, which helps to elucidate how context affects sentiment in different perspective, 4) aggregating the word-level sentiment to produce text-level sentiment. AMIC also incorporates a modified self-attention with relative positional representations, enabling the model to effectively handle nuanced linguistic complexities, such as negation.

We introduce AMIC in the case of binary classification, with a note that it can be easily extended to multiclass classification or prediction with continuous outcomes. Figure 1 presents the AMIC architecture, where $y_i$ ($i = 1, 2, ..., n$) represents the observed sentiment label of the $i^{th}$ document, $x_{ij}$ is a length-$d$ word embedding vector of the $j^{th}$ ($j = 1, 2, ..., m_i$) word in the $i^{th}$ document. Note that only $x_{ij}$ and $y_i$ are observed values and they are represented with a square box in Figure 1.

The structure of the AMIC model incentivizes the following learned behavior: $v_{ij}$ is a real-valued scalar representing the context-independent sentiment score of word $j$ in document $i$, with a positive

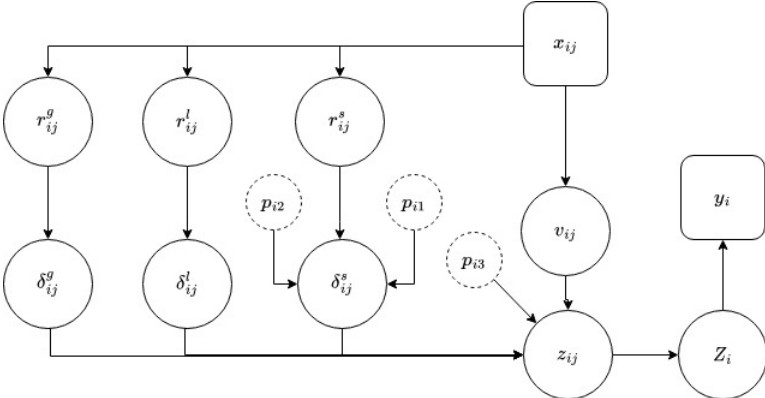

Figure 1: AMIC Architecture

value indicating positive sentiment and a negative value negative sentiment. Next $r_{ij}^s$, $r_{ij}^g$, and $r_{ij}^l$ are $d$-dimensional vectors, where $r_{ij}^s$ is optimized to learn if word $j$ in document $i$ is a sentiment or functional word. $r_{ij}^g$ learns global contextual dependency, $r_{ij}^l$ learns local contextual dependency. Then $\delta_{ij}^s$, $\delta_{ij}^g$, and $\delta_{ij}^l$ are real-valued scalars derived from $r_{ij}^s$, $r_{ij}^g$, and $r_{ij}^l$, respectively. $\delta_{ij}^s$ is the proxy indicator that takes the value of 1 or 0, where 1 indicates that the word is a sentiment word and 0 a functional word. $\delta_{ij}^g$ and $\delta_{ij}^l$ are used as global and local sentiment shifters, respectively. The value of $\delta_{ij}^g$ ranges from 0 to 10, quantifying the influence of global contextual dependency for the sentiment of the word. $\delta_{ij}^l$ represents the degree of local contextual dependency on the sentiment of the word. It takes a value between -1 and 1, where a negative value indicates the negation of the context-independent sentiment of the word in the document. Finally, $p_{i1}$, $p_{i2}$, and $p_{i3}$ are the penalty terms, $z_{ij}$ is the word's context-dependent sentiment, and $Z_i$ represents the overall sentiment score of document $i$. All $\delta$ variables are latent and learned only from the observed document level sentiment. For the remainder of this section, we provide a detailed explanation of how AMIC is constructed and trained.

We first use a three-layer feedforward operation on $x_{ij}$ to calculate $v_{ij}$, where $x_{ij}$ is a pre-trained word embedding. Thus $v_{ij}$ is context-independent, as it is not influenced by other words in the document. Next we update $r_{ij}^g$ and $r_{ij}^s$. Since $r_{ij}^g$ contains the word representations on global dependency and $r_{ij}^s$ on the identification of sentiment words, they are under the influence of document context but not affected by the order of words. Consequently, they are updated by the self-attention mechanism.

Recall that $r_{ij}^l$ represents local contextual dependency which is based on the relationships between neighboring words in a document. The self-attention with relative positional representations based on Shaw et al. (2018) follows:

$$e_{ijk} = (x_{ij}^Q)(x_{ik}^K + a_{j \to k}^K)^T, \quad \alpha_{ijk} = \frac{\exp(e_{ijk})}{\sum_{h=1}^{m_i} \exp(e_{ijh})}, \quad r_{ij}^l = \sum_{k=1}^{m_i} \alpha_{ijk}(x_{ik}^V + a_{j \to k}^V), \quad (1)$$

where the superscript $Q,K,V$ represent the Query, Key, and Value vectors, respectively; $a_{j \to k}^K$ and $a_{j \to k}^V$ are the relative positional embeddings for the Key and Value vectors, respectively; $\alpha_{ijk}$ is the attention weight incorporating relative positional representations. We propose a novel variation of the algorithm, where $e_{ijk}$ and $\alpha_{ijk}$ remain the same, but $r_{ij}^l$ becomes:

$$r_{ij}^l = \sum_{k=1}^{m_i} \alpha_{ijk}(a_{j \to k}^V). \quad (2)$$

In Equation (1) $r_{ij}^l$ is a weighted representation of the Value vector modified by the positional information. The Value vector is commonly regarded as encodings of the semantic meaning of words in the input sequence. They capture the fundamental semantic information that plays a crucial role in comprehending the entire sequence, thus proving valuable for various downstream tasks

like sentiment analysis (Tabinda Kokab et al., 2022), machine translation (Ghader & Monz, 2017), and question-answering (Seonwoo et al., 2020). Following Equation (2), $r_{ij}^l$ now is a weighted representation of the positional embedding. This adjustment allows $r_{ij}^l$ to focus on the positional information rather than encoding the semantic meaning carried by the Value vectors. By removing the Value vectors, the model becomes more sensitive to the relative positions of words, enabling it to better capture the local contextual dependencies based on their positions in the sequence. Why can AMIC allow $r_{ij}^l$ to only use the positional embedding? It is because AMIC has decomposed the semantic/sentiment information of words in multiple latent dimensions: context-independent score and context-dependent score, where the transition from the former to the latter is further explained by the joint function of global shifter and local shifter, so that $r_{ij}^l$ does not need a comprehensive word representation but a specific word representation focusing on local positional dependency. In addition, by removing the Value vectors in Equation (1), the proposed algorithm yields a more efficient model that requires roughly 30% fewer parameters compared to (Shaw et al., 2018).

Next, $\delta_{ij}^s$, $\delta_{ij}^g$, and $\delta_{ij}^l$ are updated as follows, where $\sigma$ is the sigmoid activation:

$$\delta_{ij}^s = \sigma(r_{ij}^s{}^T b^s), \qquad \delta_{ij}^g = 10 \times \sigma(r_{ij}^g{}^T b^g), \qquad \delta_{ij}^l = \tanh(r_{ij}^l{}^T b^l) \qquad (3)$$

Note that the result from the continuous activation function of $\delta_{ij}^s$ will produce an 0/1 indicator value with the constraints added by the penalty terms (explained later). A scaling factor of 10 is applied to the sigmoid function for the global sentiment shifter $\delta_{ij}^g$ to address the potential vanishing gradient problem. The hyperbolic tangent function is used to update the local sentiment shifter $\delta_{ij}^l$ which has an output range between -1 and 1.

The context-dependent sentiment score $z_{ij}$ is obtained by multiplying together the context-independent sentiment $v_{ij}$, the sentiment word indicator $\delta_{ij}^s$, the global sentiment shifter $\delta_{ij}^g$, and the local sentiment shifter $\delta_{ij}^l$:

$$z_{ij} = v_{ij} \times \delta_{ij}^s \times \delta_{ij}^g \times \delta_{ij}^l \qquad (4)$$

The product of these factors captures their collective impact on the final sentiment score for the word. If a word is identified as a function word (i.e., $r_{ij}^s = 0$), then $z_{ij} = 0$, which means that it does not contribute to the document-level sentiment score $Z_i$. For a sentiment word (i.e., $r_{ij}^s = 1$), the context-dependent score $z_{ij}$ stems from the context-independent score $v_{ij}$ modified by its global contextual dependency and local contextual dependency. The document-level sentiment score $Z_i$ is then determined by averaging the sentiment scores in the document:

$$Z_i = \frac{\sum_{j=1}^{m_i} z_{ij}}{\sum_{j=1}^{m_i} r_{ij}^s}, \qquad \hat{y}_i = \mathbf{1}_{[0.5,1]}(\sigma(Z_i)) \qquad (5)$$

where $\hat{y}_i$ denotes the predicted sentiment label. $\hat{y}_i = 1$ if $\sigma(Z_i) \geq 0.5$ and $\hat{y}_i = 0$ otherwise. The penalty terms are constructed as follows,

$$p_{i1} = c_1 \sum_{j=1}^{m_i} \sqrt{\delta_{ij}^s(1 - \delta_{ij}^s)}, \qquad p_{i2} = c_2 \sum_{j=1}^{m_i} \delta_{ij}^s, \qquad p_{i3} = c_3 \sqrt{\sum_{j=1}^{m_i} (v_{ij} \times \delta_{ij}^g \times \delta_{ij}^l)^2} \qquad (6)$$

Note that an 0/1 indicator function is not differentiable, so we can not use an exact 0/1 indicator for sentiment word identification when using backpropagation. Instead we employ a proxy indicator, $\delta_{ij}^s$, which is a differentiable sigmoid function, and we add the penalty term $p_{i1}$ to ensure that $\delta_{ij}^s$, after rounding to a certain decimal place, has a dichotomous outcome to adequately approximate an 0/1 indicator function. The function in penalty $p_{1i}$ has a dome shaped curve, encouraging $\delta_{ij}^s$ to take values close to 0 or 1.

The second penalty term $p_{2i}$ (an L1-norm) promotes sparsity in the identification of sentiment words. Our preliminary examination of the application datasets shows that the sentiment words typically account for less than 30% of all the words in a document. This observation initiates the introduction of sparsity in sentiment word identification. The third penalty term $p_{3i}$ is to ensure the stability in the estimation of $z_{ij}$ in Equation (4). Specifically, it imposes an L2 penalty to prevent the model from arbitrarily inflating the magnitude of $v_{ij} \times \delta_{ij}^g \times \delta_{ij}^l$ in situations where $\delta_{ij}^s$ may take close-to-zero values in the early training stage. $c_1$, $c_2$, and $c_3$ are tuning parameters in the penalty terms.

The parameters in AMIC are trained using gradient descent to minimize the binary cross-entropy loss and the three penalty terms:

$$l(X_i, y_i) = -\frac{1}{n} \sum_{i=1}^{n} \left( [y_i \log(\sigma(Z_i)) + (1 - y_i) \log(1 - \sigma(Z_i))] + p_{i1} + p_{i2} + p_{i3} \right). \quad (7)$$

The cross-entropy loss encourages the model to produce accurate document-level sentiment prediction. The training scheme is summarized in Algorithm 1, shown in the appendix.

## 4 RESULTS

We have evaluated AMIC on two datasets: a wine review dataset (Katumullage et al., 2022) and a Twitter Sentiment140 dataset (Go et al., 2009). The wine review dataset consists of 141,409 reviews collected from the website of the renowned wine magazine *Wine Spectator* dated from 2005 to 2016. Each year, the magazine's editors chose more than 15,000 wines for blind tasting, where they provided tasting notes, numeric ratings, and recommendations. The tasting scores are on a 100-point scale. The majority of wines have a rating in the range of 80–100. For demonstration purposes, we labeled the sentiment of a wine as positive if its rating is at least 90, and negative otherwise.

The Sentiment140 dataset consists of 1.6 million tweets with brief messages each limited to a maximum of 140 characters. The tweets collected were posted between April 6 and June 25 in 2009. Manually labeling such a large dataset would be impractical due to its size. To overcome this challenge, Go et al. (2009) adopted a technique introduced in (Read, 2005), where emoticons were utilized as sentiment labels. Out of the 1.6 million tweets, 800,000 were associated with a negative sentiment and the other 800,000 with a positive sentiment.

### 4.1 DOCUMENT LEVEL PERFORMANCE EVALUATION

For the wine dataset, we choose to use the 300-dimensional word embeddings (Glove-300-Wiki) trained on Wikipedia as the embeddings of $x_{ij}$. Glove-300-Wiki is considered to be a reliable word embedding choice for texts using formal language because Wikipedia mainly consists of documents written in formal language using proper words. The language in the wine review dataset is also standard, which makes it appropriate to use the Glove-300-Wiki embeddings. Words that are not present in the Glove-300-Wiki vocabulary were removed, resulting in an elimination of 3.3% of the words. For the Sentiment140 dataset, we employ word2vec (Mikolov et al. (2013a)) to generate word embeddings. The tweets in Sentiment140 were often written in informal language, so employing word2vec to generate wording embeddings allows AMIC to potentially use better word representation in Twitter Sentiment140.

The training-validation-test split follows a 18:1:1 ratio in both datasets. The objective of this evaluation is to compare AMIC to a number of commonly used SA methods. To avoid overlap, we have included the WCSA illustration for the wine dataset in the appendix, the effectiveness of WCSA will be fully explained using the Sentiment140 dataset application. Based on Table 1, AMIC has

Table 1: AMIC classification performance on wine and twitter datasets

| Model | Accuracy (% ), Wine | Accuracy (%), Twitter | # of Parameters |
|---|---|---|---|
| Naïve Bayes | 85.53 | 77.45 | <60k |
| Logistic Regression | 87.50 | 78.40 | <60k |
| CNN | 88.02 | 79.33 | <870k |
| BiLSTM | 88.69 | 80.21 | <100k |
| AMIC (Ours) | 88.98 | 83.73 | <1M |
| BERT | 89.12 | 86.72 | 110M |

achieved the second highest accuracy sentiment classification for wine review (0.8898), trailing marginally behind BERT (0.8912), but outperforming BiLSTM (0.8869) and CNN (0.8802). AMIC,

which is developed to conduct interpretable WCSA, does not compromise on its performance in the document-level sentiment classification of wine reviews. Interestingly, logistic regression performs well on wine reviews, despite its simplicity. This may indicate that a limited vocabulary of strong sentiment words is able to also capture global sentiment, motivating us to further explore AMIC on a more nuanced dataset. On the Sentiment140 dataset, the performance of AMIC is considerably higher than most algorithms, but lags behind BERT. We hypothesize that the informal language in this dataset might require additional tuning or additional transformer layers in order to better describe the sentiment. Even so, AMIC has far fewer parameters compared to BERT but achieves excellent performance.

## 4.2 MODEL ABLATION EVALUATION

In order to better understand the various impact of the model parameters, we perform an ablation study with the $v_{ij}$ and $\delta$ parameters, as shown in Table 2. The removal of the local sentiment shifter degrades performance most substantially, indicating the importance of local patterns recognition. This is further substantiated as all of the top performing AMIC model ablations include the local sentiment shifter. As hypothesized, this element allows negation and is a crucial aspect to the model. The sentiment indicator tends to be the next most important aspect of the model while the global sentiment shifter is least crucial. In fact, even without the global shifter's amplification, AMIC has similar performance. Having established the importance of these architectural elements, we now focus our attention on how explainable the AMIC architecture is.

Table 2: AMIC ablation study on Sentiment140 Twitter dataset

| Models | Accuracy (%) | $v_{ij}$ | Global Sentiment Shifter | Local Sentiment Shifter | Sentiment word Indicator |
|---|---|---|---|---|---|
| Ablations | 79.54 | ✓ | ✗ | ✗ | ✗ |
| | 79.63 | ✓ | ✗ | ✗ | ✓ |
| | 81.32 | ✓ | ✓ | ✗ | ✗ |
| | 81.67 | ✓ | ✓ | ✗ | ✓ |
| | 82.34 | ✓ | ✗ | ✓ | ✗ |
| | 82.54 | ✓ | ✓ | ✓ | ✗ |
| | 83.32 | ✓ | ✗ | ✓ | ✓ |
| Full Model | 83.73 | ✓ | ✓ | ✓ | ✓ |

## 4.3 INTERPRETABILITY: WCSA PERFORMANCE OF AMIC

In this section, we will examine the performance of AMIC in conducting WCSA, focusing on explaining how it is capable of providing interpretable SA results. Specifically, we give detailed description on AMIC's analysis of Sentence I and II mentioned in Section 2.2 and a number of other examples in the Sentiment140 dataset. These examples are used to illustrate AMIC's proficiency in providing informative sentiment estimation and its effectiveness in handling delicate lingistic complexities such as negation and sentiment intensifier by incorporating word positional information.

We start with Sentence I (see Table 3), which consists of two clauses seperated by a comma. In the sentiment in the first clause is positive. In the second clause, the subject "the overall experience" is modified "not bad". Although "bad" itself carries a negative sentiment, it is negated by "not", resulting in a positive sentiment too.

Table 3 provides a summary of the components used in the calculation of the context-dependent sentiment of individual words and the document-level sentiment label for Sentence I. The $v_{ij}$ column contains the context-independent sentiment score, which remains constant regardless of the context. For instance, the context-independent sentiment of "good" is 21.2 in all the sentences it appears. AMIC identifies "good", "not", and "bad" as sentiment words (i.e., $\delta_{ij}^s = 1$). AMIC effectively

Table 3: AMIC's Analysis Result of Sentence I

| Raw text | The service of the restaurant is good, the overall experience is not bad. | | | | | | | | | | | | |
|---|---|---|---|---|---|---|---|---|---|---|---|---|---|
| Input text | the | service | of | the | restaurant | is | good | the | overall | experience | is | not | bad |
| $v_{ij}$ | 7.10 | -1.0 | 6.9 | 7.1 | 19.3 | -2.1 | 21.2 | 7.1 | 30.1 | 11.4 | -2.1 | -17.4 | -28.1 |
| $\delta^s_{ij}$ | 0 | 0 | 0 | 0 | 0 | 0 | 1 | 0 | 0 | 0 | 0 | 1 | 1 |
| $\delta^g_{ij}$ | - | - | - | - | | - | 1.47 | - | - | - | - | 6.6 | 2.6 |
| $\delta^l_{ij}$ | - | - | - | - | | - | 0.45 | - | - | - | - | -0.9 | -0.8 |
| $Z_{ij}$ | 0 | 0 | 0 | 0 | 0 | 0 | 14.2 | 0 | 0 | 0 | 0 | 103.8 | 62.5 |
| $Z_i$ | 60.17 | Sentiment Label: Positive | | | | | | | | | | | |

handles negation by recognizing "bad" in the sentence is part of "not bad." The negative context-independent sentiment of "bad", -28.1, after being negated by "not", has a negative local shifter, -0.8, resulting in a positive context-dependent sentiment of 62.5. The document-level sentiment score is 60.17, indicating an overall positive sentiment conveyed in Sentence I.

Sentence II, which has the same collection of words as Sentence I, also consists of two clauses (see Table 4). The only difference is the location of the word "not". Both clauses express a negative sentiment. AMIC has identified three sentiment words in Sentence II: "not", "good", and "bad". In the first clause, the positive context-independent sentiment score of "good" (21.2, the same value as in Sentence I) is reversed by "not" in the phrase "not good" leading to a negative context-dependent sentiment score of -30.7. In the second clause, the sentiment of "bad" is not reversed because there are no negation words nearby, resulting in a context-dependent sentiment score of -11.7. The document-level sentiment score is -44.37, indicating an overall negative sentiment conveyed in Sentence II.

Table 4: AMIC's Analysis Result of Sentence II

| Raw text | The service of the restaurant is not good, the overall experience is bad. | | | | | | | | | | | | |
|---|---|---|---|---|---|---|---|---|---|---|---|---|---|
| Input text | the | service | of | the | restaurant | is | not | good | the | overall | experience | is | bad |
| $v_{ij}$ | 7.10 | -1.0 | 6.9 | 7.1 | 19.3 | -2.1 | -17.4 | 21.2 | 7.1 | 30.1 | 11.4 | -2.1 | -28.1 |
| $\delta^s_{ij}$ | 0 | 0 | 0 | 0 | 0 | 0 | 1 | 1 | 0 | 0 | 0 | 0 | 1 |
| $\delta^g_{ij}$ | - | - | - | - | | - | 6.59 | 1.46 | - | - | - | - | 2.66 |
| $\delta^l_{ij}$ | - | - | - | - | | - | 0.79 | -0.99 | - | - | - | - | 0.16 |
| $Z_{ij}$ | 0 | 0 | 0 | 0 | 0 | 0 | -91.7 | -30.7 | 0 | 0 | 0 | 0 | -11.7 |
| $Z_i$ | -44.37 | Sentiment Label: Negative | | | | | | | | | | | |

It is also interesting to point out the different treatment of "not" in these two sentences by AMIC. Note that 'not" has the same content-independent sentiment score of $-17.4$ (a negative sentiment) in both cases. This makes sense because "not" is typically used to express negation, denial, refusal, or prohibition. In Sentence I, "not" is placed next to "bad". Though both words carry a context-independent negative sentiment, together "not bad" conveys a positive sentiment. Thus, the sentiment polarity of "not" is flipped to a positive sentiment with 103.8 as its context-dependent sentiment score. In Sentence II, the sentiment of "not" is not flipped as it is positioned adjacent to "good". It is even strengthened to have a more negative context-dependent sentiment score of -91.7, capturing the clear negative sentiment expressed in the phrase "not good". This comparison further demonstrates the capability AMIC in providing nuanced understanding and accurate identification of sentiment in sentences by incorporating word position information.

Natural language has a rich representation of negative expressions, where negation can be classified into different groups in different ways (Xiang et al. (2014)), for example, preceding negation vs. succeeding negation by the location of the negation word with respect to the negated concept, or explicit negation vs. implicit negation by whether negation is in the asserted meaning or in the non-asserted content. The negation structure in both Sentence I and Sentence II is considered to be preceding negation, because the negation word "not" precedes the word whose sentiment is negated by it. It also falls into the category of explicit negation because of the use of an explicit negation word "not". The following example tweet demonstrates the model's ability to capture succeeding negation with the word "free" (Table 5).

Table 5: AMIC's Analysis Result of a Succeeding Negation Example

| Raw text | Celebrating Phil being one year cancer free! | | | | | | |
|---|---|---|---|---|---|---|---|
| Input text | celebriting | phil | being | one | year | cancer | free |
| $v_{ij}$ | 32.9 | 17.3 | -11.5 | 2.3 | -4.6 | -38.6 | 19.9 |
| $\delta_{ij}^s$ | 1 | 1 | 0 | 0 | 0 | 1 | 1 |
| $\delta_{ij}^g$ | 1.98 | 1.63 | - | - | - | 1.4 | 2.03 |
| $\delta_{ij}^l$ | 0.73 | 0.79 | - | - | - | -0.28 | 0.45 |
| $z_{ij}$ | 47.80 | 22.5 | 0 | 0 | 0 | 14.9 | 18.5 |
| $Z_i$ | 25.93 | Sentiment Label: Positive | | | | | |

AMIC recognizes "celebrating", "phil", "cancer", and "free" as sentiment words, among which only "cancer" conveys a negative context-independent sentiment, while the other three words carry a positive context-independent sentiment. The succeeding negation word "free" negates the sentiment conveyed by "cancer". Consequently, the context-independent sentiment score of "cancer" at -38.6 is shifted to a positive context-dependent sentiment score of 14.9, accurately interpreting the positive sentiment conveyed by the phrase "cancer-free". In contrast, the context-independent sentiment scores of "celebrating", "phil", and "free" remain unchanged, contributing to the overall positive sentiment conveyed in the tweet.

Next we present an example of implicit negation (Table 6). The first portion of the sentence, "Chicago was awesome", expresses a positive sentiment, whereas the second portion, "my dreams were shattered", conveys a strong negative sentiment.

Table 6: AMIC's Analysis Result of an Implicit Negation Example

| Raw text | Chicago was awesome although my dreams were shattered. | | | | | | | |
|---|---|---|---|---|---|---|---|---|
| Input text | chicago | was | awesome | although | my | dreams | were | shattered |
| $v_{ij}$ | 1.7 | -3.1 | 35.1 | 2.1 | -7.4 | 11.3 | -5.6 | -25.5 |
| $\delta_{ij}^s$ | 0 | 0 | 1 | 1 | 0 | 1 | 0 | 1 |
| $\delta_{ij}^g$ | - | - | 1.30 | 0.96 | - | 4.25 | - | 1.31 |
| $\delta_{ij}^l$ | - | - | 0.04 | 0.87 | - | -0.43 | - | 0.50 |
| $z_{ij}$ | 0.0 | 0.0 | 1.7 | 1.7 | 0 | -20.9 | 0.0 | -16.8 |
| $Z_i$ | -8.57 | Sentiment Label: Negative | | | | | | |

AMIC identifies "awesome", "although", "dreams", and "shattered" as sentiment words in the sentence. The phrase "my dreams were shattered" contains implicit negation, as it conveys a negative sentiment without using explicit negation words like "not" or "never". The implicit negation is implied through the word "shattered". Because of it, the word "dreams" which carries a positive context-independent sentiment (11.3), is coupled with a negative local sentiment shifter, resulting in a negative context-dependent sentiment (-20.9). The strong negative sentiment in the second part of the sentence leads to a overall negative document-level sentiment.

## 5 CONCLUSION

We present AMIC, an interpretable transformer model designed for sentiment classification. We show that the performance of AMIC is on par with BERT-based models, but with far fewer parameters. Moreover, we show that the most important aspects of AMIC are the local sentiment shifter, allowing interpretable negation, and the sentiment indicator, allowing neutral words to be excluded. Numerous examples demonstrate AMIC's interpretability with varying language complexity. They all illustrate AMIC's capability of conducting fine-grained WCSA by incorporating word positional information, providing detailed interpretation of analysis process, and producing correct classification results in SA. It is worth noting that AMIC achieves these goals without requiring word-level sentiment for training. Additional WCSA qualitative results of AMIC are included in the appendix.

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

# A  APPENDIX

## A.1  AMIC TRAINING ALGORITHM

To keep things concise, we represent the parameters pertaining to the computation of the context-independent sentiment score $v$ by the symbol $\theta_v$, encompassing the weight matrices for feed-forward layers. Similarly, the parameters associated with the calculation of the local sentiment shifter $\delta^l$ are denoted as $\theta_l$, which includes the parameters for the self-attention transformation, the relative positional embedding, and feedforward layers. Likewise, the parameters related to the calculation of the global sentiment shifter $\delta^g$ and the sentiment word indicator $\delta^s$ are referred to as $\theta_g$ and $\theta_s$, respectively, encompassing the parameters for the self-attention transformation and feedforward layers.

A two-pass training approach is utilized, with the first pass primarily focusing on updating $\theta_v$, and in the second pass, attention is shifted towards updating the remaining parameters. This two-pass training strategy is implemented to address potential identifiability issues, as negative values can be assumed by both $v_{ij}$ and $\delta_{ij}$. However, the objective is to ensure that positive $v_{ij}$ corresponds exclusively to positive sentiment, and vice versa. Additionally, a minibatch training scheme is employed, with the convergence condition defined as being met when the average validation loss does not display improvement in two consecutive epochs. For brevity, the indices for the minibatches have been omitted. The training scheme for AMIC is presented in Algorithm 1 below.

---

**Algorithm 1:** AMIC training procedure

**Data:** $x_{ij}, y_i$, where $i = 1, ..., n$, and $j = 1, ..., m_i$
Initialization: $\theta_v, \theta_l, \theta_g, \theta_s$
First Pass (with $z = v$):
1  **while** *not converged* **do**
2      Draw random mini batch from data
3      **for all** $(x_{ij}, y_i)$ *in batch* **do**
4         Evaluate the objective function, $l_{ce}(x_{ij}, y_i)$
5         $\theta_v \leftarrow \text{ADAM}(\nabla_{\theta_v}, l_{ce}(x_{ij}, y_i), \theta_v)$
6      **end**
7  **end**
    Second Pass (with $z = v \times \delta^s \times \delta^g \times \delta^l$);
8  set $q = 3$; $c_1 = 1e - 4$; $c_2 = 1e - 3$; $c_3 = 1e - 4$;
9  **while** *not converged* **do**
10      Draw random mini batch from data
11      **for all** $(x_{ij}, y_i)$ *in batch* **do**
12         Calculate $\delta^g_{ij}, \delta^l_{ij}, \delta^s_{ij}$
13         Evaluate the objective function, $l_{ce}(x_{ij}, y_i)$ and penalties, $p_{i1}, p_{i2}, p_{i3}$
14         $\theta_l \leftarrow \text{ADAM}(\nabla_{\theta_l}, l_{ce}(x_{ij}, y_i) + p_{i3}, \theta_l)$
15         $\theta_g \leftarrow \text{ADAM}(\nabla_{\theta_g}, l_{ce}(x_{ij}, y_i) + p_{i3}, \theta_g)$
16         $\theta_s \leftarrow \text{ADAM}(\nabla_{\theta_s}, l_{ce}(x_{ij}, y_i) + p_{i1} + p_{i2}, \theta_s)$
17      **end**
18  **end**

---

## A.2  WCSA OF THE WINE REVIEW DATASET

Table 7 presents AMIC's top 50 words with the highest sentiment scores in the wine review dataset, where font size is proportional to the sentiment score. It is less obvious why some of the words in the list convey a positive sentiment compared to the other words. Table 8 provides a few examples of how these words, which are not typically associated with positive sentiment in everyday language, convey a positive connotation in the domain of wine reviews. For instance, the adjective "stained" implies a deeper and more complex flavor profile, which is typically considered to be of high quality. Similarly, "carpet" and "fabric" are associated with wines with a rich velvety texture that is often indicative of high quality wines. The word "cascading" introduces an additional flavor profile —

suggesting that a wine has a complex and layered flavor profile. Lastly, "cognac" refers to a specific type of brandy, produced in the Cognac region of France, which is renowned for its complex flavor profile and smooth finish, making it a desirable association for high quality wines.

Table 7: AMIC'S Top 50 List of Positive Sentiment Words

| gorgeous | beautiful | ethereal | beautifully | gloriously |
|---|---|---|---|---|
| gorgeously | thoroughly | drips | beauty | impeccable |
| amazingly | exquisite | strikingly | sumptuous | cognac |
| burgundy | breed | velvet | cascading | haunting |
| seductive | finely | stuffed | soak | lovely |
| soaked | perfectly | deliciously | brilliantly | impeccably |
| wonderful | drip | luxuriant | glistening | silk |
| truffle | charms | brunello | soothing | carpet |
| champagne | perfume | seductively | fabric | unobtrusive |
| sings | swirl | stained | wonderfully | elegance |

Note: size of a word is proportional to its sentiment score

Table 8: Illustration of Examples on Less-obvious Positive Sentiment Words

| word | review texts |
|---|---|
| stained | ...the long charcoal **stained** finish has a nice tug of roasted bay leaf and truffle, and this shows terrific range... |
| | ...floral notes creamy mouthfeel and a long fruit **stained** finish... |
| carpet | like a persian **carpet** this lovely elegant champagne seamlessly weaves together its elements with fine grained texture and vibrant acidity... |
| | ...sail across a **carpet** of superfine tannins lingering on the spicy finish... |
| cascading | ...with smoky cherry and red plum flavors **cascading** along a sleek frame a charming wine... |
| | ...and then ends with a **cascading** mix of fruit mineral cedar sage... |
| fabric | ...with fine tannins woven delicately into its **fabric**... |
| | ...like a suit cut from beautiful **fabric** balanced with a vibrancy and silkiness... |
| cognac | ...rich and expressive with spun honey bergamot graphite and pastry dough notes leading to plum tart smoked almond and **cognac** flavors... |
| | ...marzipan fennel seed and **cognac** notes accent this rich and creamy blanc de blancs... |

Table 9 presents the bottom 50 words with the lowest sentiment scores. Table 10 presents a few examples featuring words that are commonly associated with positive sentiment in everyday language but are indicative of negative sentiment in wine reviews. Our analysis of the positive sentiment words reveals that fine wines are generally associated with sophistication, depth, and complexity. Consequently, terms that suggest the opposite qualities, such as "quick" or "breezy," are deemed undesirable in wine reviews. These terms imply a lack of complexity and evolution. The word "hearted" is associated with the phrase "light-hearted". The expression "light-hearted" indicates an undesirable attribute, as it suggests a wine that is too simple and lacks character. Finally, terms such as "straightforward" and "easygoing" also indicate a lack of sophistication and depth, making them descriptors for less desirable wines.

Table 9: AMIC'S Bottom 50 List of Negative Sentiment Words

| quick | generic | hearted | simple | canned |
|---|---|---|---|---|
| uncomplicated | diluted | tinny | neutral | straightforward |
| stale | cocktail | easygoing | fizzy | picnic |
| flat | lovage | greenish | unfocused | breezy |
| beaujolais | metallic | dull | easy | tail |
| modestly | decent | fade | scallion | modest |
| cucumber | cloying | watermelon | soft | parsley |
| asparagus | kosher | muddled | herbal | lemonade |
| detract | weedy | blunt | tired | muscadet |
| grass | grassy | chilled | trim | overripe |

Note: size of a word is proportional to it sentiment score

Table 10: Illustration of Examples on Less-obvious Negative Sentiment Words

| word | review texts |
|---|---|
| quick | ...light with modest citrus and green apple hints clean **quick** finish... |
| | light and **quick** with lemon pulp and jicama notes... |
| hearted | ...a friendly off dry style with light **hearted** candied lime peel and peach notes... |
| | ...light **hearted** and floral with nice melon and kiwi flavors that bounce through... |
| straightforward | ...a **straightforward** white with light notes of gala orange and basil easy to drink but loses some intensity towards the finish... |
| | ..this **straightforward** red shows light cherry herbal and vanilla flavors over ... light tannins |
| easygoing | ...bright cherry and berry notes are up front in this **easygoing** supple red... |
| | ...light and **easygoing** with pretty pear and green melon flavors... |
| breezy | ...tender with modest green apple and green melon notes featuring an open **breezy** finish... |
| | ...pear and floral notes that stay nicely defined through the **breezy** finish... |

## A.3 ADDITIONAL WCSA EXAMPLE IN THE SENTIMENT140 DATASET

In addition to negation handling, AMIC can also deal with other types of language complexity, such as use of intensifiers, which are adverbs or adverbial phrases that strengthen the meaning of other expressions and show emphasis. Table 11 shows such an example. It has two short phrases, "bad" and "very bad." Apparently, "very bad" delivers a stronger negative sentiment than just "bad". The global sentiment shifter, without bearing positional awareness, can not recognize that "very" only intensifies the second "bad", and it takes the same value of 7.61 for both occurrences of "bad" in the sentence. However, the local sentiment shifter, equipped with positional awareness, accurately recognizes that "very" is a function word that only intensifies the sentiment of the second "bad" but not the first "bad". As a result, the local sentiment shifter for the second "bad" is larger that for the first "bad", resulting in a stronger negative sentiment for the second "bad" correctly.

Table 11: AMIC Analysis Result of an Intensifier Word

| Raw text | Bad, very bad! | | |
|---|---|---|---|
| Input text | bad | very | bad |
| $v_{ij}$ | -28.1 | 3.8 | -28.1 |
| $\delta_{ij}^s$ | 1 | 0 | 1 |
| $\delta_{ij}^g$ | 7.61 | - | 7.61 |
| $\delta_{ij}^l$ | 0.67 | - | 0.92 |
| $z_{ij}$ | -142.6 | 0.0 | -196.7 |
| $Z_i$ | -169.7 | Sentiment Label: Negative | |

