# OpenReview forum: "Interpretable word-level context-based sentiment analysis"
_ICLR.cc/2024/Conference — Submitted to ICLR 2024_

### Official Review · Reviewer_7yoE · 2023-10-28

**Soundness:** 2 fair
**Presentation:** 3 good
**Contribution:** 1 poor
**Rating:** 3
**Confidence:** 4

**Summary:**

This paper proposes an attention-based multiple-instance classification model for sentiment analysis to add the interpretability of the model from word-level structure. The overall structure is well organized. The experiments show the interpretable ability of the proposed model.

**Strengths:**

The proposed model improves the interpretability from the word-level perspective and uses the MIC module. The overall structure is well organized, and the experimental results illustrate the effectiveness of the proposed model on two public SA datasets.

**Weaknesses:**

The novelty and contributions are limited. There are some existing methods using the MIC and self-attention mechanism on SA tasks. The authors should introduce them and explain the main differences as well as advantages. Chapter 4 in "Sentiment Lexicon Induction and Interpretable Multiple-instance Learning in Financial Markets". In addition, for the word level interpretability of the SA model, does the author consider combining other level information in the model as illustrated in "A Multi-Grained Self-Interpretable Symbolic-Neural Model For Single/Multi-Labeled Text Classification" -- "there is only word-level attribution but no high-level attribution such as those over phrases and clauses. Take sentiment analysis as an example, in addition to the ability to recognize the sentiment of sentences, an ideal interpretable model should be able to identify the sentiment and polarity reversal at the levels of words, phrases, and clauses."

For the AMIC architecture, the reason why using a three-layer feedforward operation should be introduced. And why not use a Transformer-based model and only use a self-attention mechanism?

For the word embedding, what are the differences between Glove-300-Wiki  and  word2vec? The author should add more comparative experiments to show the different performances and BERT embedding.
For Table 2, it is better to add more ablation studies on different datasets.
Moreover, sentiment analysis is a common NLP task, so the authors should add other baselines.

For interpretability, do the authors consider an attention map to show the performance of the attention mechanism? to illustrate the effectiveness of the proposed model?

More sentence case studies could be added in the appendix.

**Questions:**

See above.

---

### Official Review · Reviewer_UzwB · 2023-10-31

**Soundness:** 2 fair
**Presentation:** 2 fair
**Contribution:** 2 fair
**Rating:** 3
**Confidence:** 3

**Summary:**

This paper introduces AMIC, an attention-based multiple instance classification model for interpretable word-level sentiment analysis using document-level sentiment labels. AMIC offers enhanced interpretability compared to other models while maintaining competitive document-level performance. The model incorporates self-attention and positional encoded self-attention within a transparent architecture, providing insights into context-driven sentiment and decision-making.

**Strengths:**

-	Reasonable method.
-	Evaluation of two datasets with relatively detailed experimental analysis.

**Weaknesses:**

-	The techniques employed in this study are rather conventional, and in terms of text interpretability, while they do offer some assistance, their significance is limited. Furthermore, for longer sentences or sentence-level tasks (commonly addressed using BERT-based models for text encoding), they lack scalability.

-	The work presented in this paper is quite mundane, and employing large models like ChatGPT for word-level interpretability might even yield better results, enhancing its overall scalability as well.

**Text:** \
The service of the restaurant is good, the overall experience is not bad
**Prompt:** \
What is the emotion of this sentence, analyzed at the word level
**ChatGPT 3.5:** \
At the word level, the emotion of this sentence can be broken down as follows:

"good" implies a positive emotion.
"not bad" implies a somewhat positive or neutral emotion. While it contains a negation ("not"), it's a double negative, and when used colloquially, it often means "quite good" or "satisfactory."

So, the overall emotion at the word level is generally positive, with a touch of reservation or neutrality.

-	The related work should be updated with more recent related works.

-	Figure 1 appears quite blurry. I recommend redrawing the figure to ensure it meets the required resolution of 300 DPI.

-	The formulas in the article appear quite unusual and are not conducive to understanding, especially when there are three formulas in a single line. It's not recommended.

-	The format of the references in the paper needs to be consistent. If a referenced paper has already been published, it should not be cited in the arXiv format. This should be updated to reflect the appropriate citation style for the published version.

-	Writing errors are common across the overall paper. Examples could be found in “Typos, Grammar, Style, and Presentation Improvements”.

**Typos, Grammar, Style, and Presentation Improvements:**

-	On page 1, “… weighting, indication of …” => “… weighting, an indication of …”
-	On page 1, “… insights on how context influences …” => “… insights into how context influences …”
-	On page 1, “… multiple instance classification model …” => “… multiple instance classification models …”
-	On page 1, “… methods which focus …” => “… methods that focus …”
-	On page 2, “… support vector machine …” => “… support vector machines …”
-	On page 2 “… long-short term …” => “… long-short-term …”
-	On page 2 “… shown BERT …” => “… shown that BERT …”
-	On page 2 “… in text can …” => “… in the text can …”
-	On page 3 “… sentiment score at …” => “… sentiment scores at …”
-	On page 3 “… help a SA model …” => “… help an SA model …”
-	On page 3 “… its ignorance to the …” => “… its ignorance of the …”
-	On page 6 “… the sentiment of a wine …” => “… the sentiment of wine …”
-	On page 6 “… sentiment and the other …” => “… sentiment, and the other …”
-	On page 6 “… follows a 18:1:1 ratio …” => “… follows an 18:1:1 ratio …”
-	On page 7 “… has similar performance …” => “… has a similar performance …”
-	On page 7 “… is able to also capture …” => “… is able to capture …”
-	On page 7 “… various impact of …” => “… various impacts of …”
-	On page 7 “… local patterns recognition.” => “… local pattern recognition.”
-	On page 7 “… aspect to the …” => “… aspect of the …”
-	On page 7 “… delicate lingistic complexities …” => “… delicate linguistic complexities …”
-	On page 7 “… clauses seperated by …” => “… clauses separated by …”
-	On page 8 “… the capability AMIC in provid …” => “… has a similar
-	On page 9 “… providing detailed interpretation …” => “… providing a detailed interpretation …”
-	On page 9 “… of analysis process …” => “… of the analysis process …”

**Questions:**

Please see the Weaknesses.

---

### Official Review · Reviewer_kPxy · 2023-11-07

**Soundness:** 2 fair
**Presentation:** 2 fair
**Contribution:** 1 poor
**Rating:** 3
**Confidence:** 4

**Summary:**

This paper presents AMIC, an innovative attention-based multiple-instance classification approach designed to conduct sentiment analysis at the word level while leveraging document-level sentiment labels for interpretability. The model's architecture employs both self-attention mechanisms and positional encodings, enhancing its transparency and allowing for a clear understanding of how word context influences sentiment determination.

**Strengths:**

The model's integration of the MIC module enhances word-level interpretability, and its efficacy is demonstrated through experiments conducted on two public sentiment analysis datasets.

**Weaknesses:**

The critique highlights the perceived lack of innovation and the limited contributions of the study, suggesting that the methods used are standard and offer minimal advancements in text interpretability. It points out a scalability issue with longer text sequences. It suggests that utilizing more advanced language models for sentence-level tasks or even LLMs (like ChatGPT or LLaMa) for word-level interpretability could improve performance and scalability significantly.

In addressing the use of different word embeddings, the question is about the rationale behind using GloVe-300-Wiki for the wine dataset and word2vec for the Sentiment140 dataset. The author is encouraged to explain the reasons for this choice and to provide a comparison between the two embeddings to elucidate their differences and justify their specific applications within the study.

The related work section is weak and does not cover SOTA models. It needs to be updated with more recent studies.

Figure 1 is noticeably unclear and requires enhancement for better visibility.

The article has many writing errors throughout and needs to be thoroughly corrected.

**Questions:**

Please see the weaknesses section.

---

### Meta-Review · Area_Chair_kZEb · 2023-12-09

**Metareview:**

In this paper, the authors introduce AMIC, an interpretable transformer model designed for sentiment classification. The authors show that the performance of AMIC is on par with BERT-based models but with far fewer parameters. AMIC also acts as the local sentiment shifter,
allowing interpretable negation, and the sentiment indicator, allowing neutral words to be excluded. The experimental results show the effectiveness of the proposed method.

However, the reviewers have many concerns about the paper but the authors did not respond to any of them. I recommend to reject the paper.

**Justification For Why Not Higher Score:**

The authors did not respond to the reviewers' concerns

**Justification For Why Not Lower Score:**

N/A

---

### Decision · Program_Chairs · 2024-01-16

Reject